# Monte Carlo Planning with Large Language Model for Text-Based Game Agents

**Zijing Shi**
AAII, University of Technology Sydney
zijing.shi@student.uts.edu.au

**Meng Fang**
University of Liverpool
Meng.Fang@liverpool.ac.uk

**Ling Chen**
AAII, University of Technology Sydney
ling.chen@uts.edu.au

## Abstract

Text-based games provide valuable environments for language-based autonomous agents. However, planning-then-learning paradigms, such as those combining Monte Carlo Tree Search (MCTS) and reinforcement learning (RL), are notably time-consuming due to extensive iterations. Additionally, these algorithms perform uncertainty-driven exploration but lack language understanding and reasoning abilities. In this paper, we introduce the Monte Carlo planning with Dynamic Memory-guided Large language model (MC-DML) algorithm. MC-DML leverages the language understanding and reasoning capabilities of Large Language Models (LLMs) alongside the exploratory advantages of tree search algorithms. Specifically, we enhance LLMs with in-trial and cross-trial memory mechanisms, enabling them to learn from past experiences and dynamically adjust action evaluations during planning. We conduct experiments on a series of text-based games from the Jericho benchmark. Our results demonstrate that the MC-DML algorithm significantly enhances performance across various games at the initial planning phase, outperforming strong contemporary methods that require multiple iterations. This demonstrates the effectiveness of our algorithm, paving the way for more efficient language-grounded planning in complex environments. [1]

## 1 Introduction

Text-based games serve as valuable environments for studying various natural language processing (NLP) and sequential decision-making problems (Narasimhan et al., 2015; Xu et al., 2020). In these games, agents navigate environments using textual commands and deal with limited observability. Unlike simple synthetic games (Côté et al., 2019), human-designed adventure games feature dynamic state spaces and sparse rewards, presenting significant challenges (Hausknecht et al., 2020). Existing game agents are typically based on reinforcement learning (RL) and employ $\epsilon$-greedy or softmax policies for action exploration. However, they lack long-term planning abilities (Osborne et al., 2022).

Planning for text-based games presents unique challenges, as each language action is treated as a discrete token, making uncertainty-driven exploration without understanding the potential future impacts or the natural language described game state. Previous works that integrate Monte Carlo Tree Search (MCTS) with learning models have shown remarkable proficiency in classical games such as Go and Atari (Browne et al., 2012; Świechowski et al., 2023). These methods are based on the architectures pioneered by AlphaGo and AlphaGo Zero, which utilize policy and value networks to evaluate and prioritize potential moves, thereby significantly enhancing gameplay (Silver et al., 2016; 2017). However, MCTS still faces challenges in real-world scenarios. Learning models typically need a substantial warm-up period to learn effectively. Their performance relies on data obtained from MCTS planning, which is confined by the practical limitations of tree size and depth.

---

[1]Our code is available at https://textgamer.github.io/mc-dml/.

Specifically, in text-based games, MCTS lacks the necessary language understanding and reasoning abilities. In response, Jang et al. (2020) suggested guiding exploration in MCTS planning through the evaluation of action's semantic similarity. This approach assumes that an action may possess value if its similar actions taken previously have high $Q$ values. While effective in certain games, this assumption may be less reliable in more dynamic settings.

The recent emergence of Large Language Models (LLMs) have shown remarkable capabilities in quickly generating viable initial plans for decision-making tasks, even with minimal or no prior examples. Some research explores various prompting techniques, such as Reflection (Shinn et al., 2024) and Tree-of-Thought (Yao et al., 2024), which further enhance LLM reasoning for interactive tasks. Despite achieving near-saturated performance in simpler environments such as ALFworld (Shridhar et al., 2020), they continue to face challenges in more complex settings. A primary challenge lies in translating the plans generated by LLMs into executable actions. Furthermore, LLMs often struggle to balance exploration and exploitation, which hinders their ability to navigate extensive state spaces efficiently.

In this study, we explore the potential of LLMs to enhance MCTS planning in complex interactive tasks. We aim to answer two questions: (1) Can LLMs, with their encoded knowledge, enhance action exploration within MCTS planning, thereby improving sample efficiency and task performance? (2) Can LLMs, with their few-shot learning capabilities, dynamically adapt action guidance based on past experiences during planning? To address these questions, we introduce the Monte Carlo planning with Dynamic Memory-guided Large language model (MC-DML). This algorithm leverages the language understanding and commonsense reasoning capabilities of LLMs and the exploration benefits of tree-search approaches. By integrating both in-trial and cross-trial memory into LLMs, MC-DML enables dynamic adjustments of action evaluation during the MCTS planning.

We conduct experiments using a series of text-based games from the Jericho benchmark (Hausknecht et al., 2020). These games are characterized by numerous branching paths and sparse rewards. The agent, operating under limited observability, must extensively explore the environment to solve complex puzzles. Our results demonstrate that the MC-DML algorithm enhances performance across various games at the initial planning phase, outperforming strong contemporary methods that require multiple iterations for policy optimization. Additionally, we perform ablation studies to highlight the role of the memory mechanism in LLM policy.

Our main contributions are summarized as follows: First, we propose an MCTS-based algorithm that integrates an LLM to enhance action exploration in complex textual interactive tasks. Second, we develop an LLM agent equipped with both in-trial and cross-trial memory, enabling dynamic language action value estimation in tree-search planning phrase. Third, our experiments on a series of text-based games demonstrate that the proposed algorithm significantly improves performance across multiple games at the initial planning phase, outperforming strong contemporary methods that require multiple iterations.

## 2 PRELIMINARY

### 2.1 MONTE-CARLO TREE SEARCH

**Upper Confidence bound for Trees (UCT)** MCTS (Kocsis & Szepesvári, 2006; Coulom, 2006) operates by iteratively developing a decision tree through four key phases: selection, expansion, simulation, and backpropagation. Within the search tree, the standard MCTS method uses UCT to choose action $a^*$ at each node, balancing exploitation based on the $Q$-value with exploration driven by uncertainty. The formula for UCT is as follows:

$$a^* = \arg \max_{a \in \mathcal{A}(s)} \left[ Q(s,a) + C_{uct} \cdot \sqrt{\frac{\ln N(s)}{N(s,a)}} \right] \tag{1}$$

where $Q(s,a)$ is the average reward for action $a$ in state $s$, $N(s,a)$ is the number of times action $a$ is chosen in state $s$, $N(s)$ is the visit count to state $s$, $C_{uct}$ is a constant that balances exploration and exploitation.

**Predictor UCT (PUCT)** One limitation of UCT is its dependence on Monte Carlo averages to estimate state values, which can result in significant search inefficiencies, especially in text-based

games with high branching factors. PUCT partially overcomes these challenges by incorporating PUCB, which utilizes a prior action distribution $\pi(\cdot|s)$ to estimate action values under state $s$ and prioritize exploration (Silver et al., 2017; 2018). The formula for PUCT is as follows:

$$a^* = \arg \max_{a \in \mathcal{A}(s)} \left[ Q(s,a) + C_{puct} \cdot \pi(a|s) \cdot \frac{\sqrt{N(s)}}{1 + N(s,a)} \right] \tag{2}$$

Here, $\pi(\cdot|s)$ is usually a neural network that trained via behavioral cloning using $(s, a^*)$ samples from previous tree search results. However, PUCT still faces several challenges. Firstly, the training data for policy network is sourced from MCTS planning. In practice, due to limitations in tree size and depth, the training data may not be optimal. To achieve effective learning, it often requires multiple iterations using a planning-then-learning paradigm, which is time-consuming. Additionally, this training data is specific to certain environments, limiting the ability of the policy network to generalize across various games. Furthermore, the nature of its supervised learning approach restricts its strategic depth and impairs its long-term planning capabilities.

## 2.2 TEXT-BASED GAMES AS POMDPS

The text-based game can be modeled as a Partially Observable Markov Decision Process (POMDP) (Narasimhan et al., 2015), represented by $(\mathcal{S}, T, \mathcal{A}, \mathcal{O}, R, \gamma)$. At each time $t$, the agent cannot directly observe the environmental state $s_t$ from $\mathcal{S}$. Instead, it infers the state through a textual observation $o_t$ from $\mathcal{O}$. When the agent performs an action $a_t$ from $\mathcal{A}$, the environment transitions to the next state according to the hidden transition function $T$. Meanwhile, the agent receives a reward $r_t = R(s_t, a_t)$ and the subsequent observation $o_{t+1}$ from the game environment. The agent's goal is to optimize actions to maximize the expected total discounted rewards $R_t = \mathbb{E}[\sum_{t=0}^{\infty} \gamma^t r_t]$, where $\gamma$ ranges from 0 to 1, indicating the discount factor.

## 3 METHOD

In this study, we focus on human-designed text adventure games, which present two significant challenges. First, these games feature a vast combinatorial action space. To manage this complexity, benchmarks such as Jericho provide a predefined set of valid actions at each step by filtering out inadmissible commands. However, this still results in a dynamic action space that varies with the game state, leading to numerous game branches. Moreover, these games are characterized by sparse rewards and multiple bottleneck states. Figure 1 illustrates an example of a bottleneck state in the game `Zork1`. Based on the current observation, the agent selects from the available actions, leading to different game branches. An agent optimized for cumulative rewards might choose `open trapdoor`, resulting in a significant immediate reward but also leading to subsequent death. To progress in the game, the agent must explore necessary actions without receiving any immediate reward signals. This requires the agent to combine semantic reasoning with long-term planning capabilities.

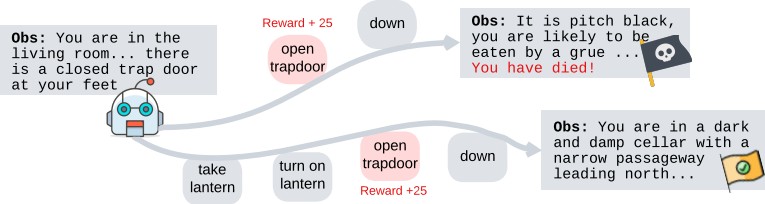

Figure 1: An example bottleneck state from the game `Zork1`.

To address these challenges and the limitations of current MCTS-based algorithms, we introduce the MC-DML algorithm. We provide a comprehensive introduction to MC-DML in Section 3.1, outline the algorithm during a single planning process in Section 3.2, and discuss the innovative aspects of MC-DML in Section 3.3.

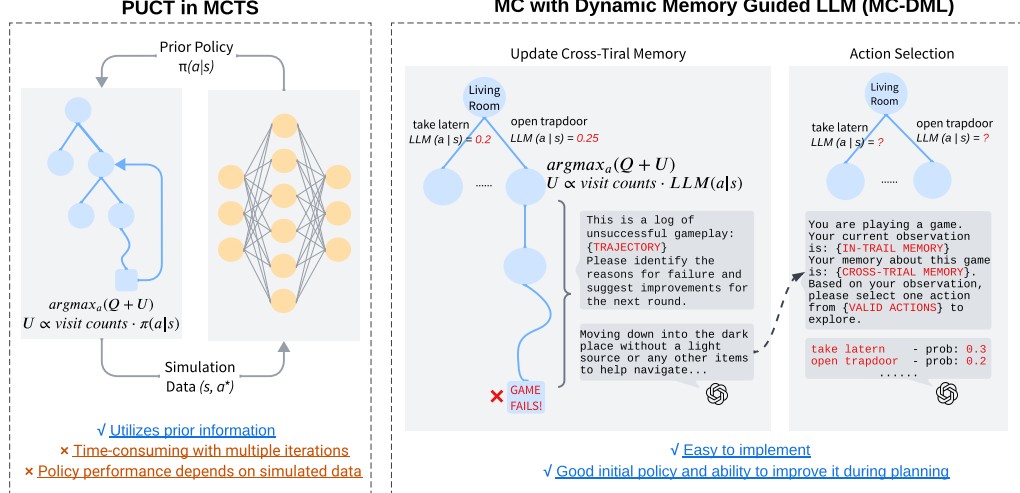

Figure 2: A comparison of the PUCT and MC-DML algorithms. PUCT trains its policy through imitation learning from self-play data. In contrast, MC-DML uses a LLM as the initial policy. During planning, the LLM learns from past failure trajectories and adjusts the action value estimates. This approach more closely aligns with the human thought process.

### 3.1 MONTE CARLO PLANNING WITH DYNAMIC MEMORY-GUIDED LLMS (MC-DML)

MC-DML consists of four stages: selection, expansion, simulation, and backpropagation, and finally predicts an action based on the simulations. During the expansion phase, MC-DML employs an LLM as the prior policy within the PUCT algorithm. Based on the current tree state, the LLM assigns non-uniform search priorities to each optional action. After an action is selected, the expansion phase adds a new node to the search tree. In the simulation phase, MC-DML conducts multiple rollouts from this new node to evaluate the potential outcomes of the chosen action. The simulation results are then backpropagated to update the Q-value estimates and visit counts of the relevant nodes. In text-based games, the current state is not fully observable. Therefore, we equip the LLM with a dynamic memory mechanism, utilizing in-trial memory $\mathcal{M}_i$ and cross-trial memory $\mathcal{M}_c$. $\mathcal{M}_i$ contains the current trajectory history, representing the game state, while $\mathcal{M}_c$ includes experiences from previous failure trajectories, used to dynamically adjust the action value estimation.

**Learning from In-Trial Memory** The in-trial memory $\mathcal{M}_i$ includes a sequence of past observations and actions. Using this memory, we prompt the LLM to generate a probability distribution of valid actions $\pi(\cdot|s)$ grounded in commonsense. The probability of an action $a$ is calculated by accumulating the conditional probabilities of its tokens. We use the GPT-3.5 model, which provides log probabilities for the top potential answers. These probabilities are then normalized using the softmax function. For APIs where token probabilities are unavailable, this can be achieved through self-consistency (Wang et al., 2022) and verbalized methods (Lin et al., 2022) [2].

**Reflection on Cross-Trial Memory** In-trial memory is a form of short-term memory that relies on the LLM's commonsense but lacks flexibility. Inspired by Shinn et al. (2024), we develop cross-trial memory $\mathcal{M}_c$, an interpretable and enduring form of episodic memory that allows agents to learn from past failures. In MCTS, the agent repeatedly simulates from the root node to explore various paths. This restart mechanism allows agent to reflection on the segment of trajectory and resume play from that save point. Figure 2 illustrates the process of updating and utilizing cross-trial memory in MC-DML. During the tree search, when the agent encounters a terminal state due to game failure, the LLM analyzes this trajectory and generates a reflection. In subsequent simulations

---

[2]Self-consistency methods estimate the probability of an answer by sampling multiple responses from the LLM. Verbalized methods leverage a well-designed prompt to instruct the LLM to output the most likely answers along with their corresponding probabilities.

under the same root node, we combine in-trial memory and cross-trial memory to adjust the LLM's action estimations. The formula for action selection in MC-DML is as follows:

$$a^* = \arg\max_{a \in \mathcal{A}} \left[ Q(s,a) + C_{puct} \cdot LLM(a|\mathcal{M}_i, \mathcal{M}_c, p) \cdot \frac{\sqrt{N(s)}}{1 + N(s,a)} \right] \tag{3}$$

where $LLM(a|\mathcal{M}_i, \mathcal{M}_c, p)$ is the action probability distribution generated by the LLM policy, $p$ is the given prompt consisting of an instruction and optional actions $\mathcal{A}$, $Q(o,a)$ is the average reward for action $a$ in tree state $s$.

## 3.2 ALGORITHM

---

**Algorithm 1** Monte Carlo Planning with Dynamic Memory-Guided LLM (MC-DML)

---

1: **procedure** SEARCH($s_0$)
2:     $s_0 \leftarrow O(s_0)$
3:     $h_0 \leftarrow o_0$
4:     **repeat**
5:         SIMULATE($s_0, h_0, 0$)
6:     **until** MaxDepthReached()
7:     **return** $\arg\max_{a \in \mathcal{A}} Q(h_0, a)$
8: **end procedure**
9: **procedure** SIMULATE($s, h, t$)
10:     **if** GAMEFAIL($s$) **then**
11:         reflection $\leftarrow$ LLM($h, p_{\text{reflection}}$)
12:         $\mathcal{M}_c \leftarrow \mathcal{M}_c + $ reflection
13:         **return** 0
14:     **end if**
15:     **if** $t = $ planning horizon $H$ **then**
16:         **return** 0
17:     **end if**
18:     $[a, \text{rollout}] \leftarrow$ SELECTACTION($h$)
19:     $[r, s', o'] \leftarrow \mathcal{T}(s, a)$
20:     $h' \leftarrow h + a + o'$
21:     **if** rollout **then**
22:         $R' \leftarrow$ ROLLOUT($s', h', t+1$)
23:     **else**
24:         $R' \leftarrow$ SIMULATE($s', h', t+1$)
25:     **end if**
26:     $R \leftarrow r + \gamma \cdot R'$
27:     $N(h) \leftarrow N(h) + 1$
28:     $N(h, a) \leftarrow N(h, a) + 1$
29:     $Q(h, a) \leftarrow Q(h, a) + \frac{R - Q(h,a)}{N(h,a)}$
30:     **return** $R$
31: **end procedure**

32: **procedure** SELECTACTION($h$)
33:     $\mathcal{M}_i \leftarrow$ LASTPART($h$)
34:     $\pi(a|s) \leftarrow$ LLM($\mathcal{M}_i, \mathcal{M}_c, p_{\text{action\_probs}}$)
    $a^* \leftarrow \arg\max_{a \in \mathcal{A}} [Q(s,a) +$
        $c_{\text{puct}} \pi(a|s) \sqrt{\frac{N(s)}{N(s,a)+1}}$
35:     **if** $N(s,a) = 0$ **then**
36:         rollout $\leftarrow$ true
37:     **else**
38:         rollout $\leftarrow$ false
39:     **end if**
40:     **return** $[a^*, \text{rollout}]$
41: **end procedure**
42: **procedure** ROLLOUT($h, s, t$)
43:     **if** GAMEFAIL($s$) **then**
44:         reflection $\leftarrow$ LLM($h, p_{\text{reflection}}$)
45:         $\mathcal{M}_c \leftarrow \mathcal{M}_c + $ reflection
46:         **return** 0
47:     **end if**
48:     **if** $t = $ planning horizon $H$ **then**
49:         **return** 0
50:     **end if**
51:     $o \leftarrow O(s)$
52:     $a \sim$ Uniform($\mathcal{A}$)
53:     $[r, s', o'] \leftarrow \mathcal{T}(s, a)$
54:     **return** $r + \gamma \cdot$ ROLLOUT($s', t+1$)
55: **end procedure**

---

We now describe the operation of MC-DML in a single planning episode, as outlined in Algorithm 1. For each simulation, MC-DML initializes the root node $s_0$ and new trajectory $h_0$ to construct the state (lines 2-3). An action $a^*$ is chosen based on the $Q$ value, visit counts, and the LLM policy (lines 34-35). The LLM policy determines the action probability distribution using the in-trial memory $\mathcal{M}_i$ and the cross-trial memory $\mathcal{M}_c$. The in-trial memory $\mathcal{M}_i$ is a portion of the trajectory $h$ (line 33), while the cross-trial memory $\mathcal{M}_c$ consists of reflections generated from previous failed trajectories (lines 10-13, 44-47). MC-DML iteratively selects actions to execute and updates the visit counts and estimated $Q$ values (line 26-29). When encountering leaf nodes, it expands the tree and performs a rollout, using a uniform policy to sample actions and returning the discounted reward. Upon completion of the search process, the agent will execute an final action based on the estimated $Q$ value and receive a new observation.

### 3.3 Novelty in Comparison to Prior Algorithms

We now more explicitly discuss comparisons to a few other approaches. The approach most related to ours is LLM-MCTS, which uses an LLM as a prior policy in MCTS (Zhao et al., 2024). While the LLM can serve as a good initial policy, it cannot improve the policy based on past experience and external feedback. This makes LLM-MCTS well-suited for commonsense planning tasks, such as object rearrangement in household environments, but less effective in uncertain environments like text-based games.

For text-based games, MC-LAVE-RL is one of the SOTA methods that combines MCTS with RL while considering the semantic sharing between actions (Lee et al., 2021). During MCTS planning, an exploration reward is added to each action $a$, estimated through the $Q$-values of its semantically similar actions. This approach addressed the bottleneck state in the game `Zork1` (see Figure 1). Actions such as collecting items typically have high value in games, resulting in the action `take lantern` being assigned a higher exploration bonus than the action `open trapdoor`. However, its performance beyond games remains to be validated. In this study, rather than relying on item collection within LLM prompts, we focus on developing a more general solution. MC-DML simulates human gameplay by combining in-trial and cross-trial memory, mimicking how humans retain both recent detailed information and significant past experiences. We avoid introducing any prior game knowledge or human-designed hints in the LLM prompts.

## 4 Experiments

### 4.1 Experimental Setup

**Environments**   We conduct experiments on 9 text-based games provided by the Jericho benchmark (Hausknecht et al., 2020). These games provide a variety of challenges such as darkness, nonstandard actions, inventory management, and dialog Among them, `Zork1`, `Deephome`, and `Ludicorp` are categorized as difficult games, as their optimal completion paths require over 300 steps, with an average of more than 14 available actions per step. The remaining games are categorized as possible games (Hausknecht et al., 2020). At each step $t$, the observation from the Jericho game engine includes a description of the state. This is augmented with location and inventory information by issuing the "look" and "inventory" commands, forming $o_t$. Additionally, we utilize the valid action handicap provided by Jericho. For further analysis of these games, refer to Appendix A.

**Implementation Details**   For the LLM policy, we use `gpt-3.5-turbo-0125` as the backend model with a sampling temperature set to 0. We query the LLM for the index of the optimal action and retrieve the log probabilities for the top 20 tokens at that index. For absent actions, we assign a log probability of -10. These log probabilities are then normalized using softmax with a temperature of 5. The in-trial memory is defined as $(o_{t-1}, a_{t-1}, o_t)$, and the size of the cross-trial memory $k$ is set to 3. We allow each root node to store up to $k$ reflections. If the number of reflections exceeds $k$, cross-memory collection is terminated early due to the sufficient experiences. When necessary, we compress the input information using a truncation function to ensure that it fits within the LLM's input window constraints.

For the tree search component, we adopt a dynamic pruning strategy; the search depth is dynamically adjusted between a minimum depth $d_{\min}$ and a maximum depth $d_{\max}$. The algorithm begins with $d_{\min}$. If the highest Q-value of the selected action node is 0, the depth is increased by $\Delta d$ and the search is repeated, up to $d_{\max}$. This setting takes into account the uneven distribution of steps with rewards in the game. We provide a statistical analysis of the game's step distribution in Appendix A and an ablation study of this setting in Section 4.3. Further details about the experimental implementation can be found in Appendix B.

**Evaluation**   We first evaluate the performance of MC-DML in comparison with baseline methods on a series of text-based games. Next, we compare MC-DML with the intermediate scores of MCTS-based baselines during multiple iterations in the game `Zork1`. Then, we conduct ablation studies on a subset of games to evaluate the importance of different memory mechanisms in MC-DML. Finally, we provide further qualitative analysis, including how MC-DML addresses bottleneck states in the game `Zork1`.

**Baseline** We consider 10 baselines, including RL-based agents, LLM-based agents, and MCTS-based agents. All these baselines except MC!Q*BERT assume access to valid actions from the Jericho benchmark. Among these, PUCT-RL and MC-LAVE-RL algorithms serve as direct comparators to MC-DML. (1) **DRRN** (He et al., 2015) : The Deep Reinforcement Relevance Network (DRRN) is an RL-based agent that utilizes a Q-based softmax policy. This policy is parameterized with GRU encoders and decoders and is trained using the Temporal-Difference (TD) loss. (2) **KG-A2C-Hard** (Ammanabrolu & Hausknecht, 2020): An actor-critic method using a knowledge graph for state representation, with a hard valid action constraint. (3) **MC!Q*BERT** (Ammanabrolu et al., 2020): An extension of KG-A2C that leverages BERT for knowledge graph construction and includes knowledge-graph-based intrinsic rewards. (4) **MPRC-DQN** (Guo et al., 2020): A DQN-based method that formulates action selection as a multi-paragraph reading comprehension task, retrieving relevant historical observations and jointly encoding them with the current state for Q-value prediction. (5) **RC-DQN** (Guo et al., 2020): A DQN-based agent that applies a reading comprehension model to compute Q-values from the current observation alone, without multi-paragraph retrieval. (6) **BiKE+CBR** (Atzeni et al., 2021): A knowledge-graph-based A2C agent augmented with a case-based reasoning framework that explicitly stores and reuses successful experiences to improve out-of-distribution generalization. (7) **LLM agent**: We employ the LLM directly as the agent to interact with the environment, aiming to assess potential data contamination within the LLM. In this setting, the LLM selects actions from the valid action set based on the current trajectory history complete the game. We set the temperature of the LLM to 0 and select the action with the highest output probability. (8) **Reflection agent** (Shinn et al., 2024): We allow the LLM to perform reflection, which is then used to guide its interactions with the environment in the next round. (9) **PUCT-RL** (Jang et al., 2020): PUCT-RL uses PUCT as a policy improvement operator for DRRN, alternating between PUCT planning and supervised learning of self-generated actions. (10) **MC-LAVE-RL** (Jang et al., 2020): MC-LAVE is one of the SOTA models on the Jericho benchmark that combines MCTS with RL while considering the semantic sharing between actions.

## 4.2 MAIN RESULTS

Tables 1 and 2 present the performance of MC-DML compared with RL-based baselines and LLM/MCTS-based baselines, respectively, across a set of 9 games. We observe the following key findings. First, when compared to RL-based baselines, MC-DML achieves better or comparable results in 6 out of 9 games. In comparison to LLM/MCTS-based baselines, MC-DML outperforms or matches their performance in 8 out of 9 games. Secondly, in challenging games like `Deephome`, MC-DML overcomes multiple bottlenecks, achieving nearly double the performance of MC-LAVE-RL. In possible games like `Pentari` and `Detective`, MC-DML even completes the games fully. In other possible games, such as `Library` and `Temple`, it also approaches the best possible score within the given number of steps. Finally, the LLM policy performs poorly, likely due to its inability to balance exploration and exploitation. This also indicates that LLM does not have knowledge of the game's walkthrough under the current prompting setting.

| Algorithms / Games | DRRN | KG-A2C-Hard | MC!Q*BERT | MPRC-DQN | RC-DQN | BiKE + CBR | MC-DML |
|---|---|---|---|---|---|---|---|
| Zork1 | 32.6 | 40.2 ± 0.4 | 41.6 | 38.3 | 38.8 | 44.3 | **48.66 ± 1.89** |
| Deephome | 1 | 20 ± 2.1 | 8 | 1 | 1 | 1 | **67 ± 1.41** |
| Ludicorp | 13.8 | 19.8 ± 1.0 | 22.8 | 19.7 | 17 | **23.8** | 19.67 ± 1.7 |
| Pentari | 27.2 | 44 ± 0.9 | 58 | 44.4 | 43.8 | 52.1 | **70 ± 0.0** |
| Detective | 197.8 | 338 ± 3.4 | 330 | 317.7 | 291.3 | 326.1 | **346.67 ± 9.43** |
| Library | 17 | 17 ± 0.0 | 19 | 17.7 | 18.1 | **22.3** | 21 ± 0.0 |
| Balances | 10 | 10 | 10 | 10 | 10 | **11.9** | 10 ± 0.0 |
| Temple | 7.4 | **8 ± 0.0** | **8** | **8** | **8** | 7.8 | **8 ± 0.0** |
| Ztuu | 21.6 | 5 ± 0.0 | 11.8 | - | - | - | 23.67 ± 1.9 |

Table 1: Comparison of MC-DML with RL-based baselines on text-based games from Jericho benchmark. MC-DML indicate averages over 3 independent runs. We omit the results of some baselines on the `ztuu` game because they revealed unbounded reward loops (Guo et al., 2020).

| Algorithms Games | LLM-based | | MCTS-based | | Ours |
|---|---|---|---|---|---|
| | LLM | Reflection | PUCT-RL | MC-LAVE-RL | MC-DML |
| Zork1 | 0 | 5 | 38.2 ± 0.8 | 45.2 ± 1.2 | **48.66 ± 1.89** |
| Deephome | 1 | 1 | 28.6 ± 2.9 | 35 ± 0.6 | **67 ± 1.41** |
| Ludicorp | 4 | 4 | 18 ± 0.0 | **22.8 ± 0.2** | 19.67 ± 1.7 |
| Pentari | 5 | 5 | 64 | 68 | **70 ± 0.0** |
| Detective | 30 | 30 | 322 ± 2.0 | 330 ± 0.0 | **346.67 ± 9.43** |
| Library | 6 | 6 | 19 ± 0.0 | 19 ± 0.0 | **21 ± 0.0** |
| Balances | **10** | **10** | **10 ± 0.0** | **10 ± 0.0** | **10 ± 0.0** |
| Temple | **8** | **8** | **8 ± 0.0** | **8 ± 0.0** | **8 ± 0.0** |
| Ztuu | 0 | 5 | 5 ± 0.0 | 7 ± 2.7 | **23.67 ± 1.9** |

Table 2: Comparison of MC-DML with LLM-based and MCTS-based baselines on Jericho benchmark text-based games.

Table 3 shows the results of MC-DML alongside the intermediate scores of the MCTS-based baselines during multiple iterations on the game `Zork1`. We would like to emphasize that for the PUCT-RL and MC-LAVE-RL algorithms, the final result is computed based on the policy and $Q$-function obtained after 4 iterations, which is when convergence is reached. In each iteration, these algorithms conducted 25 independent planning sessions to collect trajectories and experience replay for policy learning. Unlike these approaches, MC-DML does not require a planning-then-learning paradigm. It can adjust the initial policy and estimated action values guided by dynamic memory.

| | PUCT-RL | | MC-LAVE-RL | | MC-DML |
|---|---|---|---|---|---|
| | Tree Search | RL | Tree Search | RL | Tree Search |
| Iteration 1 | 31.9 ± 1.4 | 36.6 ± 1.0 | 30.4 ± 2.0 | 36.6 ± 1.0 | **48.66 ± 1.89** |
| Iteration 2 | 35.8 ± 0.0 | 37.4 ± 1.0 | 36.1 ± 0.1 | 38.2 ± 0.8 | *N/A* |
| Iteration 3 | 35.3 ± 0.2 | 39.0 ± 0.0 | 41.2 ± 0.5 | 43.0 ± 1.0 | *N/A* |
| Iteration 4 | 35.2 ± 0.4 | 38.2 ± 0.8 | 43.8 ± 0.1 | 45.2 ± 1.2 | *N/A* |

Table 3: Experimental results of MC-DML with the intermediate scores of the MCTS-based baseline during multiple iterations on the game `Zork1`. Our MC-DML achieves superior results with its initial planning.

## 4.3 ABLATION STUDIES

To evaluate the importance of the memory mechanisms and dynamic pruning strategy in MC-DML, we conduct several ablation studies on a subset of games. We compare the performance of MC-DML without dynamic pruning (DP), without $\mathcal{M}_c$, without DP, and without $\mathcal{M}_c$ and DP, $\mathcal{M}_c$, and $\mathcal{M}_i$. When disregarding the DP, we follow the experimental setup of Jang et al. (2020), using a fixed search depth for each game. Without $\mathcal{M}_c$, the LLM's action estimates are based on the current historical trajectory. Without both $\mathcal{M}_c$ and $\mathcal{M}_i$, the LLM's action estimates at time $t$ rely only on the current state node $o_t$. The results show that using DP significantly improves performance in the game `Zutt`, but has little effect on other games. Removing $\mathcal{M}_c$ reduces game scores, and removing both $\mathcal{M}_c$ and $\mathcal{M}_i$ results in an even larger drop in scores, highlighting the importance of these memory mechanisms.

| | MC-DML | w.o. $\mathcal{M}_c$ | w.o. DP | w.o. $\mathcal{M}_c$, DP | w.o. $\mathcal{M}_c$, $\mathcal{M}_i$, DP |
|---|---|---|---|---|---|
| Zork1 | **48.66 ± 1.89** | 38.33 ± 2.89 | 48 ± 2.45 | 38 ± 5.2 | 31.67 ± 4.7 |
| Deephome | 67 ± 1.41 | 62.66 ± 0.94 | **67.4 ± 0.8** | 64.33 ± 0.94 | 51 ± 14.9 |
| Detective | **346.67 ± 9.43** | 326.67 ± 4.71 | 334 ± 4.9 | 323.33 ± 4.7 | 320 ± 0.0 |
| Ztuu | **23.67 ± 1.9** | 20.66 ± 0.47 | 7.8 ± 0.56 | 7 ± 0.81 | 6.33 ± 0.94 |

Table 4: Ablation results on a subset of games. For the ablation models, we report the average score over 3 independent runs. Overall, both the $\mathcal{M}_c$ and $\mathcal{M}_i$ are crucial to our MC-DML.

## 4.4 ANALYSIS

Table 5 presents an illustrative example of search results for MC-DML and MC-DML w.o. $\mathcal{M}_c$ on a bottleneck state in the game `Zork1`. Without the reflection module $\mathcal{M}_c$, the LLM assigns a high value to the action `open trap` due to its semantic alignment with the current state, which also provides an immediate reward. Although this action eventually leads to failure, the agent, lacking the ability to reflect on its mistakes, continues to explore it, resulting in both a high Q-value and $N(s, a)$. Consequently, the agent ends up selecting this action during the final execution step, which explains why it gets stuck in a bottleneck state. However, in MC-DML, the LLM generates a reflection based on failed trajectories and store it in the memory. The reflection might be, "Ensure you have a light source before entering dark areas," altering the action probability distribution in subsequent simulations. After sufficient exploration, the agent obtains an accurate value estimation and ultimately selects the `take lantern` action at the current state. Ultimately, the agent selects the optimal action `take lantern`, even though it does not provide any immediate reward. Similar bottleneck states are also addressed in the game `Deephome`. Additional trajectory examples of MC-DML playing `Deephome` are provided in the Appendix D.

| MC-DML | open trap | open case | take sword | take lantern | take all | east | turn on lantern |
|---|---|---|---|---|---|---|---|
| $Q(s, a)$ | 4.41 | 11.41 | 9.31 | **14.26** | 0.00 | -8.12 | -1.42 |
| $LLM(a\|\mathcal{M}_c, \mathcal{M}_i, p)$ | 0.16 | 0.13 | 0.10 | 0.22 | 0.10 | 0.08 | 0.17 |
| $N(s, a)$ | 21 | 39 | 27 | 252 | 6 | 2 | 3 |
| **w.o. $\mathcal{M}_c$** | **open trap** | **open case** | **take sword** | **take lantern** | **take all** | **east** | **turn on lantern** |
| $Q(s, a)$ | **13.02** | 9.92 | 8.38 | 12.66 | 3.17 | -1.85 | 4.73 |
| $LLM(a\|\mathcal{M}_i, p)$ | 0.24 | 0.20 | 0.21 | 0.10 | 0.10 | 0.05 | 0.06 |
| $N(s, a)$ | 176 | 36 | 72 | 34 | 17 | 5 | 10 |

Table 5: An illustrative example of search results for MC-DML and MC-DML w.o. $\mathcal{M}_c$ on a bottleneck state in the game `Zork1`. Regarding the differing scales between LLM value and $Q(s, a)$, during simulations, the LLM value is multiplied by a scale factor $C_{PUCT}$.

## 5 RELATED WORK

**Action Selection in MCTS**   MCTS-based algorithms thrives in large search spaces by selectively sampling promising actions (Osborne et al., 2022). The prevalent PUCT algorithm enhances this process by predicting action values using prior knowledge, typically obtained from historical data through imitation learning (Silver et al., 2017; 2018). Current research on MCTS is directed towards developing contextual action value estimators to further refine action exploration (Lee et al., 2021; Sztyglic et al., 2021). Specifically, in language-ground settings, Branavan et al. (2012) utilizes a multi-layer neural network to extract relevant text segments from game documents. This text is then integrated into the Monte-Carlo search framework to facilitate linguistically-informed decision-making. Jang et al. (2020) introduced MC-LAVE-RL for solving text-based games, a method that incorporates value sharing among actions during the search process. Specifically, an action is encouraged for exploration if similar actions taken previously have high $Q$-values. While effective in certain games, this assumption may be less reliable in more dynamic settings.

**Interactive Planning with LLM**   It is important to underscore our research focus. While recent studies have introduced search-based prompting approaches to enhance LLMs' reasoning capabilities by exploring generated thoughts (Yao et al., 2024; Ding et al., 2023), our study takes a distinct direction, emphasizing the large-scale planning under limited observability. In this domain, some studies utilize large LLMs as direct policies for interactive tasks, which yield interesting results but also exhibit certain limitations (Huang et al., 2022; Zhu et al., 2023; Fang et al., 2024). One such limitation is the difficulty in translating the plans created by LLMs into executable actions. Another is the inability of LLMs to balance exploration with exploitation. To address these issues, some research efforts use LLMs to formulate high-level plans and guide RL agents in performing specific actions (Shukla et al., 2023; Liu et al., 2024; Dalal et al., 2024; Zhang et al., 2024). However, these RL agents often struggle to conduct long-term planning. The study most closely aligned with

ours is Zhao et al. (2024), which employs an LLM as a fixed prior policy within MCTS to address common sense tasks. Whereas these tasks are more intuitive and can be effectively addressed by leveraging the world prior knowledge of LLMs, text-based games present greater uncertainty, thus posing significant challenges.

**Text-based Game Playing Agent**   Recent research has explored RL agents with varying architectures for solving text-based games (He et al., 2015; Narasimhan et al., 2015; Ammanabrolu & Hausknecht, 2020; Xu et al., 2021; Ryu et al., 2022; Tuyls et al., 2022; Shi et al., 2023b;a). Innovations in this field address the problem of combinatorial action spaces (Zahavy et al., 2018; Yao et al., 2020), modeling state space utilising knowledge graphs (Ammanabrolu & Hausknecht, 2020; Adhikari et al., 2020; Xu et al., 2020), integrating question-answering and reading comprehension modules (Ammanabrolu et al., 2020; Xu et al., 2022; Dambekodi et al., 2020). These agents rely on $\epsilon$-greedy or softmax policies, which restrict their capacity for long-term planning. To overcome this limitation, Jang et al. (2020) proposed MC-LAVE-RL, which integrated MCTS with RL to solve text-based games, while also considering semantic sharing of actions between nodes. Following this line, our study aims to extend the capabilities of these agents by combining MCTS with LLM, enhancing their strategic depth and adaptability in complex scenarios.

## 6   Conclusion

In this study, we propose the MC-DML algorithm. MC-DML leverages the prior knowledge embedded in LLM to guide action exploration during MCTS planning. The LLM is equipped with a dynamic memory mechanism to adjust action value estimation based on historical experience. MC-DML simulates human gameplay by mimicking how humans retain both recent detailed information and significant past experiences. Our results demonstrated that the MC-DML enhances performance across various games.

**Limitation**   We utilise an LLM for value estimation during MCTS planning that combines in-trial memory and cross-memory. However, for simplicity in our current setup, we define in-trial memory as the trajectory within a shorter time window. In these games, some puzzles may relate to clues encountered much earlier, such as a spell or an item seen long ago. This places demands on the LLM's "Needle In a Haystack" ability. Future work could explore more efficient in-trial memory storage and retrieval mechanisms.

## Acknowledgments

This project is partially supported by ARC DP240101349. We are grateful to the anonymous reviewers for their insightful feedback, which significantly enhanced our work.

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

## A    GAME STATISTICS

We conduct experiments upon 9 games provided by the Jericho Game Suite (Hausknecht et al., 2020). Different from those generated through pre-defined simple rules (Côté et al., 2019), the games we use are more complex, making them even challenging for the human players. These games have diverse themes and genres. For example, in the game "Ludicorp", the player appears to be a modern citizen being located in an office building. In another game "Zork1", the player is put into a fantasy world that she/he has to find the treasure in the mazes while escaping from the troll. Some games contain nonstandard actions (e.g., the spells), which are unlikely to be understood by the language model pre-trained with commonsense knowledge.

Table A shows the game statistics calculated from the walkthrough of each game. The Avg Actions per Step refers to the average number of valid actions available at each step of the game. The Walkthrough Length represents the minimum number of steps required to complete the game optimally, showing the shortest possible solution. The Avg Steps per Reward measures the average distance between two reward-triggering steps, reflecting how frequently rewards are distributed throughout the game. The Max Steps per Reward indicates the maximum number of steps a player might take between two rewards, highlighting the sparsest distribution of rewards. Finally, the Max Score represents the highest possible score an agent can achieve in the game.

|  | Avg Actions per Step | Walkthrough Length | Avg Step per Reward | Max Step per Reward | Max Game Score |
|---|---|---|---|---|---|
| Zork1 | 15.96 | 396 | 9.12 | 51 | 350 |
| Deephome | 19.47 | 327 | 5.90 | 53 | 300 |
| Ludicorp | 14.52 | 364 | 3.69 | 45 | 150 |
| Pentari | 5.16 | 49 | 6.43 | 16 | 70 |
| Detective | 7.16 | 51 | 1.96 | 5 | 360 |
| Library | 7.73 | 52 | 3.67 | 6 | 30 |
| Balances | 23.18 | 122 | 13.44 | 54 | 50 |
| Temple | 15.25 | 182 | 21.38 | 46 | 35 |
| Ztuu | 33.96 | 84 | 4.53 | 14 | 100 |

Table 6: Game statistics on text-based games from Jericho benchmark.

## B    IMPLEMENTATION DETAILS

**MC-DML.** We set the discount factor to 0.95 and the number of simulations to 50 multiplied by len($\mathcal{A}$). We set $C_{puct}$ to 50. Specifically, it is set to 20 for the games `Deephome` and `Library`, and to 200 for the game `Detective`. The above configuration follows the work of Jang et al. (2020). We set $d_{min}$ to 10, $d_{max}$ to 30, and the step increment $\Delta d$ to 20. This configuration allows the algorithm to start with a conservative search depth and expand progressively when necessary. The LLM policy uses `gpt-3.5-turbo-0125` as the backend model with a sampling temperature set to 0. We query the LLM for the index of the optimal action and retrieve the log probabilities for the top 20 tokens at that index. For absent actions, we assign a log probability of -10. These log probabilities are then normalized using softmax with a temperature of 5. The in-trial memory is set to $(o_{t-1}, a_{t-1}, o_t)$. The size of the cross-trial memory $K$ is set to 3.

**LLM agent.** For the LLM agent baseline, we use `gpt-3.5-turbo-0125` with a sampling temperature of 0.1. The agent selects actions based on the highest probability from the model's output. However, we observe that the agent often enters loops due to ineffective exploration. To address this, if the agent repeats the same action in the same state five consecutive times, it switches to a random action.

**Reflection agent.** For the Reflection agent baseline, we use the same settings as the LLM agent baseline but equip it with a cross-memory module, allowing the agent to learn through reflection and trial-and-error. Specifically, the LLM plays multiple rounds of the game, generating reflections at the end of each round, which are then used as input for the next round. In our experiments, we limit the number of reflections to 3.

## C  LLM PROMPTS

In this section, we provide the prompts used for action value estimation by the LLM, as well as the prompts used for reflection.

### C.1  PROMPTS FOR ACTION VALUE ESTIMATES

```
You are a player in a text-based adventure game. Your task is to
    evaluate and select actions that are promising based on the given
    context.

Your memory of playing this game previously is: {CORSS_TRIAL_MEMORY}
You are now facing the following state:
{IN_TRIAL_MEMORY}

Considering the current state and previous memories, please select the
    action most worth exploring from the following list:
{VALID_ACTIONS}
Respond by providing the index of the action only. Your response should
    be a single integer, without any extra formatting, spaces,
    punctuation, or text.
```

### C.2  PROMPTS FOR REFLECTION

```
You will receive a log of unsuccessful gameplay:
{TRAJECTORY}

Please briefly identify the reasons for failure and provide a suggestion
    for improvement in the next round. Do not summarize the gameplay
    trajectory; respond with your suggestion in a single sentence.
```

## D  TRAJECTORY EXAMPLE

```
[OBS] mining center This is the main center of all of the mining that
    takes place in Deephome. To the east are coal mines, rarely used
    because of the nature of the generator, and to the south are the ore
    mines, used for simultaneously mining all kinds of metal ores from
    the deeps of the mountains. To the north is the main street. [LOOK]
    This is the main center of all of the mining that takes place in
    Deephome. To the east are coal mines, rarely used because of the
    nature of the generator, and to the south are the ore mines, used
    for simultaneously mining all kinds of metal ores from the deeps of
    the mountains. To the north is the main street. [INV] You are
    carrying: - king's order - a warning note - a lantern (providing
    light)
[ACTION] east
[REWARD] 1
[GAME SCORE] 15

[OBS] coal mines The coal mines are rather stark, for by the time the
    dwarves moved into Deephome, they had already developed the
    technology to use the world's core for energy, what they called
    geothermal power. The rock on the wall looks like it has been dug in
    some, however. The mine opens up to the mining center to the west.
    [LOOK] coal mines The coal mines are rather stark, for by the time
    the dwarves moved into Deephome, they had already developed the
    technology to use the world's core for energy, what they called
    geothermal power. The rock on the wall looks like it has been dug in
    some, however. The mine opens up to the mining center to the west.
    [INV] You are carrying: - king's order - a warning note - a lantern
    (providing light)
```

```
[ACTION] push coal to warning
[REWARD] 10
[GAME SCORE] 25

[OBS] You open the generator furnace. [LOOK] city generator Here is a
    large wonder of technology and ingenuity. The generator runs on the
    natural heat of the world, having pipes that lead all the way to
    near the core of the world. However, it requires a small fire to
    start the process of activating the power. The generator itself is
    large and black, having a small furnace in the very front. A doorway
    leads to the west, out into the street. [INV] You are carrying: -
    king's order - a warning note - a lantern (providing light)
[ACTION] turn lever on
[REWARD] 30
[GAME SCORE] 56

[OBS] extreme south main street [LOOK] extreme south main street Only
    people working for the city actually ever come down here, unless
    they are on a tour. A few mining operations lie to the south of this
    road, and to the east is the main generator for the city. To the
    southwest, there is a doorway leading to the city water works. The
    street ends here but continues to the north. [INV] You are carrying:
    - king's order - a warning note - a lantern (providing light)
[ACTION] get in southwest
[REWARD] 1
[GAME SCORE] 57

[OBS] water works This room is filled with scores and scores of pipes,
    leading this way and that, all connecting with the walls at one
    point or another. A large wheel sticks out of a rather significant
    pipe and there is a sign above it. To the northeast is the main
    road. [LOOK] water works This room is filled with scores and scores
    of pipes, leading this way and that, all connecting with the walls
    at one point or another. A large wheel sticks out of a rather
    significant pipe and there is a sign above it. To the northeast is
    the main road. [INV] You are carrying: - king's order - a warning
    note - a lantern (providing light)
[ACTION] get in northeast
[REWARD] 0
[GAME SCORE] 57

[OBS] extreme south main street [LOOK] extreme south main street Only
    people working for the city actually ever come down here, unless
    they are on a tour. A few mining operations lie to the south of this
    road, and to the east is the main generator for the city. To the
    southwest, there is a doorway leading to the city water works. The
    street ends here, but continues to the north. [INV] You are
    carrying: - king's order - a warning note - a lantern (providing
    light)
[ACTION] north
[REWARD] 0
[GAME SCORE] 57

[OBS] intersection [LOOK] intersection Here meet the two major roads of
    the city proper: Main Street, which runs north and south, and the
    Royal Road which runs west from here. To the east is a guard post
    with a sign next to it. [INV] You are carrying: - king's order - a
    warning note - a lantern (providing light)
[ACTION] west
[REWARD] 1
[GAME SCORE] 58

[OBS] east royal road This road leads east to an intersection and west
    toward the castle of the regional lord. It is lined with high walls
    and gates that open to lesser nobles' palaces. There is a gate open
```

```
        to the south. [LOOK] east royal road This road leads east to an
        intersection and west toward the castle of the regional lord. It is
        lined with high walls and gates that open to lesser nobles' palaces.
        There is a gate open to the south. [INV] You are carrying: - king's
        order - a warning note - a lantern (providing light)
[ACTION] west
[REWARD] 1
[GAME SCORE] 59

[OBS] west royal road This road is quite beautiful, decorated on its
        sides with fluorescent mosses that feed on the minerals in the
        stones that line the sides of the roads. Somehow, the mosses do not
        leave their designated stones. High walls on both sides make the
        street feel more like a hall than an open passageway, and gates
        leading to palaces break up the monotony of the stone. A single gate
        is open to the north. The road continues east and to the west is the
        outer court of the lord's palace. [LOOK] west royal road This road
        is quite beautiful, decorated on its sides with fluorescent mosses
        that feed on the minerals in the stones that line the sides of the
        roads. Somehow, the mosses do not leave their designated stones.
        High walls on both sides make the street feel more like a hall than
        an open passageway, and gates leading to palaces break up the
        monotony of the stone. A single gate is open to the north. The road
        continues east and to the west is the outer court of the lord's
        palace. [INV] You are carrying: - king's order - a warning note - a
        lantern (providing light)
[ACTION] west
[REWARD] 1
[GAME SCORE] 60

[OBS] outer court High walls surround this court, and a long pathway
        leads from the gates to the east to the throne room to the west.
        Most of the palace is locked securely, but the courtyard is still
        open, for the very reason of allowing a dwarven reclaimer to have
        access to the throne room. [LOOK] outer court High walls surround
        this court, and a long pathway leads from the gates to the east to
        the throne room to the west. Most of the palace is locked securely,
        but the courtyard is still open, for the very reason of allowing a
        dwarven reclaimer to have access to the throne room. [INV] You are
        carrying: - king's order - a warning note - a lantern (providing
        light)
[ACTION] west
[REWARD] 1
[GAME SCORE] 61

[OBS] throne room The throne room is much more simple and quaint than
        you might expect. A large tapestry hangs behind the throne itself
        and a long red carpet leads from the great doorway to the east, and
        except for the throne itself, there is little here. You can see a
        luminous spirit here. [LOOK] throne room The throne room is much
        more simple and quaint than you might expect. A large tapestry hangs
        behind the throne itself and a long red carpet leads from the great
        doorway to the east, and except for the throne itself, there is
        little here. You can see a luminous spirit here. [INV] You are
        carrying: - king's order - a warning note - a lantern (providing
        light)
[ACTION] east
[REWARD] 0
[GAME SCORE] 61

[OBS] outer court [LOOK] outer court High walls surround this court, and
        a long pathway leads from the gates to the east to the throne room
        to the west. Most of the palace is locked securely, but the
        courtyard is still open, for the very reason of allowing a dwarven
```

```
    reclaimer to have access to the throne room. [INV] You are carrying:
    - king's order - a warning note - a lantern (providing light)
[ACTION] east
[REWARD] 0
[GAME SCORE] 61

[OBS] west royal road [LOOK] west royal road This road is quite
    beautiful, decorated on its sides with fluorescent mosses that feed
    on the minerals in the stones that line the sides of the roads.
    Somehow, the mosses do not leave their designated stones. High walls
    on both sides make the street feel more like a hall than an open
    passageway, and gates leading to palaces break up the monotony of
    the stone. A single gate is open to the north. The road continues
    east and to the west is the outer court of the lord's palace. [INV]
    You are carrying: - king's order - a warning note - a lantern
    (providing light)
[ACTION] north
[REWARD] 1
[GAME SCORE] 62

[OBS] greater noble's palace This palace rises high, probably consisting
    of ten or twelve floors, all carved out of the mountain. The walls
    are covered with beautiful murals and tapestries, the floors are
    mosaics with fountains and baths interrupting them. Many gold and
    silver objects sit here gleaming in the light. It is clearly a
    wealthy abode. To the south is the gate that leads to the main
    street. [LOOK] greater noble's palace This palace rises high,
    probably consisting of ten or twelve floors, all carved out of the
    mountain. The walls are covered with beautiful murals and
    tapestries, the floors are mosaics with fountains and baths
    interrupting them. Many gold and silver objects sit here gleaming in
    the light. It is clearly a wealthy abode. To the south is the gate
    that leads to the main street. [INV] You are carrying: - king's
    order - a warning note - a lantern (providing light)
[ACTION] south
[REWARD] 0
[GAME SCORE] 62

[OBS] west royal road [LOOK] west royal road This road is quite
    beautiful, decorated on its sides with fluorescent mosses that feed
    on the minerals in the stones that line the sides of the roads.
    Somehow, the mosses do not leave their designated stones. High walls
    on both sides make the street feel more like a hall than an open
    passageway, and gates leading to palaces break up the monotony of
    the stone. A single gate is open to the north. The road continues
    east and to the west is the outer court of the lord's palace. [INV]
    You are carrying: - king's order - a warning note - a lantern
    (providing light)
[ACTION] east
[REWARD] 0
[GAME SCORE] 62

[OBS] east royal road [LOOK] east royal road This road leads east to an
    intersection and west toward the castle of the regional lord. It is
    lined with high walls and gates that open to lesser nobles' palaces.
    There is a gate open to the south. [INV] You are carrying: - king's
    order - a warning note - a lantern (providing light)
[ACTION] south
[REWARD] 1
[GAME SCORE] 63

[OBS] lesser noble's palace This palace is quite beautiful, with many
    mosaics on the floors and several tapestries hanging on the walls.
    The columns that support the building are etched with pictures from
    dwarven lore and are exquisitely done. However, despite the
```

attention to beauty, there is little of anything of extreme cost
here, for dwarven craftsmanship is always ornate and admirable. The
gates to the wall lead north to the royal road. [LOOK] lesser
noble's palace This palace is quite beautiful, with many mosaics on
the floors and several tapestries hanging on the walls. The columns
that support the building are etched with pictures from dwarven lore
and are exquisitely done. However, despite the attention to beauty,
there is little of anything of extreme cost here, for dwarven
craftsmanship is always ornate and admirable. The gates to the wall
lead north to the royal road. [INV] You are carrying: – king's order
– a warning note – a lantern (providing light)
[ACTION] north
[REWARD] 0
[GAME SCORE] 63

[OBS] east royal road [LOOK] east royal road This road leads east to an
intersection and west toward the castle of the regional lord. It is
lined with high walls and gates that open to lesser nobles' palaces.
There is a gate open to the south. [INV] You are carrying: – king's
order – a warning note – a lantern (providing light)
[ACTION] east
[REWARD] 0
[GAME SCORE] 63

[OBS] intersection [LOOK] intersection Here meet the two major roads of
the city proper: Main Street, which runs north and south, and the
Royal Road which runs west from here. To the east is a guard post
with a sign next to it. [INV] You are carrying: – king's order – a
warning note – a lantern (providing light)
[ACTION] north
[REWARD] 0
[GAME SCORE] 63

[OBS] s. main street [LOOK] s. main street This section of the main
street is dedicated to quieter pursuits. To the east is the library,
and to the west is the great temple. Main Street runs north and
south from here. [INV] You are carrying: – king's order – a warning
note – a lantern (providing light)
[ACTION] east
[REWARD] 1
[GAME SCORE] 64

[OBS] dwarven library The library is quite large: floor after floor of
books rises before you, clinging to the walls of this circular room.
Stairs lead up the middle, spiraling along and pausing at landings
at each floor. At the very center of the building you can look
straight up and see the roof, and on it a mural. The exit is to the
west. You can see Fresto's Exorcisms and Tasty Hors d'Oeuvres and
Leshosh's Encyclopedia here. [LOOK] dwarven library The library is
quite large: floor after floor of books rises before you, clinging
to the walls of this circular room. Stairs lead up the middle,
spiraling along and pausing at landings at each floor. At the very
center of the building you can look straight up and see the roof,
and on it a mural. The exit is to the west. You can see Fresto's
Exorcisms and Tasty Hors d'Oeuvres and Leshosh's Encyclopedia here.
[INV] You are carrying: – king's order – a warning note – a lantern
(providing light)
[ACTION] west
[REWARD] 0
[GAME SCORE] 64

[OBS] s. main street [LOOK] s. main street This section of the main
street is dedicated to quieter pursuits. To the east is the library,
and to the west is the great temple. Main Street runs north and

```
      south from here. [INV] You are carrying: - king's order - a warning
      note - a lantern (providing light)
[ACTION] west
[REWARD] 1
[GAME SCORE] 65

[OBS] great temple of Kraxis The once great and shining temple of the
      one god, Kraxis, has been desecrated. Animal dung has been smeared
      on the beautiful wall murals, many of the tiles of the floor have
      been ripped up and thrown across the room, and the altar, a single
      large rock, unhewn by hands, has the statue of a pig-headed god on
      top of it. The large main doors lie to the east. You can see a dark
      spirit here. [LOOK] great temple of Kraxis The once great and
      shining temple of the one god, Kraxis, has been desecrated. Animal
      dung has been smeared on the beautiful wall murals, many of the
      tiles of the floor have been ripped up and thrown across the room,
      and the altar, a single large rock, unhewn by hands, has the statue
      of a pig-headed god on top of it. The large main doors lie to the
      east. You can see a dark spirit here. [INV] You are carrying: -
      king's order - a warning note - a lantern (providing light)
[ACTION] east
[REWARD] 0
[GAME SCORE] 65

[OBS] s. main street [LOOK] s. main street This section of the main
      street is dedicated to quieter pursuits. To the east is the library,
      and to the west is the great temple. Main Street runs north and
      south from here. [INV] You are carrying: - king's order - a warning
      note - a lantern (providing light)
[ACTION] north
[REWARD] 0
[GAME SCORE] 65

[OBS] n. main street [LOOK] n. main street This was once a bustling
      trading center with shops lining the sides of the streets, and
      smaller, less affluent merchants' booths sitting outside of the
      doorways, offering items of lower quality at a substantially lower
      rate. At this particular point in the street you see a clothier to
      the northeast, a bakery to the east, and a scrivener's to the west.
      To the north is the main hall, and the street runs further south.
      [INV] You are carrying: - king's order - a warning note - a lantern
      (providing light)
[ACTION] get in northeast
[REWARD] 1
[GAME SCORE] 66

[OBS] clothier In this room you see the tools of a clothier's trade,
      generally dwarven-shaped wooden mannequins, a table for cutting, and
      other tools scattered about. The exit is to the southwest. [LOOK]
      clothier In this room you see the tools of a clothier's trade,
      generally dwarven-shaped wooden mannequins, a table for cutting, and
      other tools scattered about. The exit is to the southwest. [INV] You
      are carrying: - king's order - a warning note - a lantern (providing
      light)
[ACTION] get in southwest
[REWARD] 0
[GAME SCORE] 66

[OBS] n. main street [LOOK] n. main street This was once a bustling
      trading center with shops lining the sides of the streets, and
      smaller, less affluent merchants' booths sitting outside of the
      doorways, offering items of lower quality at a substantially lower
      rate. At this particular point in the street you see a clothier to
      the northeast, a bakery to the east, and a scrivener's to the west.
      To the north is the main hall, and the street runs further south.
```

```
        [INV] You are carrying: - king's order - a warning note - a lantern
        (providing light)
[ACTION] west
[REWARD] 1
[GAME SCORE] 67

[OBS] scrivener's This room is very plain and has half a dozen desks and
    chairs where the scriveners would sit and write documents for those
    who could not read. The exit is to the east. [LOOK] scrivener's This
    room is very plain and has half a dozen desks and chairs where the
    scriveners would sit and write documents for those who could not
    read. The exit is to the east. [INV] You are carrying: - king's
    order - a warning note - a lantern (providing light)
[ACTION] east
[REWARD] 0
[GAME SCORE] 67

[OBS] n. main street [LOOK] n. main street This was once a bustling
    trading center with shops lining the sides of the streets, and
    smaller, less affluent merchants' booths sitting outside of the
    doorways, offering items of lower quality at a substantially lower
    rate. At this particular point in the street you see a clothier to
    the northeast, a bakery to the east, and a scrivener's to the west.
    To the north is the main hall, and the street runs further south.
    [INV] You are carrying: - king's order - a warning note - a lantern
    (providing light)
[ACTION] east
[REWARD] 1
[GAME SCORE] 68
```

