# OpenReview forum: "Monte Carlo Planning with Large Language Model for Text-Based Game Agents"
_ICLR.cc/2025/Conference — ICLR 2025 Poster_

### Official Review · Reviewer_djd9 · 2024-10-24

**Soundness:** 3
**Presentation:** 3
**Contribution:** 3
**Rating:** 8
**Confidence:** 4

**Summary:**

This paper introduces a new planning and learning algorithm called Monte Carlo with Dynamic Memory-guided Large language model (MC-DML). This method leverages the language understanding capabilities of LLMs and the exploratory advantages of tree search algorithms. The approach is similar to Predictor UCT: actions are selected based on the Q-value function plus a probability distribution over valid actions p(a|s). In this work, p(a|s) is not trained but produced by a prompted LLM. Applied to Jericho text-based games, GPT3.5 is used to generate p(a|s) at each game state based on both an in-trial memory (M_i = sequence of past observations & actions), and a cross-trial memory (M_c = a collection of LLM generated reflections based on previous failures). In addition, the Monte Carlo search used in this work applies dynamic pruning to save search time and favor exploration.
Experimental results on 9 Jericho games show that MC-DML yields better performance when compared to RL-based, LLM-based, and other MCTS-based methods. Ablation studies show the importance of the cross-trial memory M_c.

**Strengths:**

This paper proposes a novel method to combine the advantages of LLMs and tree-search algorithms. The idea of including both in-trial and cross-trial elements inside the decision-making process of choosing the next action is an interesting and effective technique that can inspire future work.
The paper is well written, the experiments correspond to the research questions, and are well motivated.

**Weaknesses:**

**W1.** Some technical aspects are not always clear, see below and the list of questions for things to clarify.

The paper could benefit from a more detailed explanation of the way each LLM prompt is constructed. In particular what is inside the {TRAJECTORY}, {IN-TRIAL MEMORY}, and {CROSS-TRIAL MEMORY} variables. An example in the appendix could help the reader understand what information the LLM has access to when making its decision.
A discussion about how the lengthy game trajectories are handled could also clarify some technical aspects.

**W2.** The Jericho benchmark has been used in many previous works, additional baselines could be included in the experimental results to make them more significant.
In particular MPRC-DQN & RC-DQN from Guo et al. [1], Bike+CBR from Atzeni et al. [2], and LDTs from Gontier et al. [3]

[1] Guo, X., Yu, M., Gao, Y., Gan, C., Campbell, M., & Chang, S. (2020). Interactive fiction game playing as multi-paragraph reading comprehension with reinforcement learning. arXiv preprint arXiv:2010.02386

[2] Atzeni, M., Dhuliawala, S., Murugesan, K., & Sachan, M. (2021). Case-based reasoning for better generalization in textual reinforcement learning. arXiv preprint arXiv:2110.08470.

[3] Gontier, N., Rodriguez, P., Laradji, I., Vazquez, D., & Pal, C. (2023). Language Decision Transformers with Exponential Tilt for Interactive Text Environments. arXiv preprint arXiv:2302.05507.

**Questions:**

Q1. Do you take into account the next step rewards (when present) during the Monte Carlo Tree Search? If so, how? If not, why not?

Q2. What is the average input token length of your two prompts? Given that they include trajectories, they may be longer than the 4191 context length limit of gpt3.5. Do you compress the trajectories in some way? Since states are made of the game state transition message + the description of the room + the description of inventory, sequences of (s, a) could become lengthy. For instance, consecutive room and inventory descriptions can be the same over multiple time steps if actions do not move the player or change its inventory.

Q3. Do you include the initial description of the game as part of the state? Often the opening message of the game will give valuable information about the goal/objective of the game.

Q4. When doing MCTS rollouts/simulations, how do you sample actions? Is it random or do you use an informed process during simulation as well?

Q5. Why did you select these 9 Jericho games? Could you report performance across all 33 games in the appendix?

---

> ### Author Response · Authors · 2024-11-22
> **Response to Review [Part 1/2]**
>
> We appreciate your recognition of our work and your valuable feedback. We hope the following responses address your concerns.
>
> > **Q1:** Some technical aspects are not always clear, see below and the list of questions for things to clarify...
>
> **A1:** Thank you for your question. Below, we provide detailed explanations of the prompt components, and specific examples will be presented in the Appendix.
>
> **Trajectory:**  We define a trajectory as the sequence of observation-action pairs from the initial root node up to a terminal node in the search tree. Formally, it is represented as: $\tau = \left( o_0, a_0, o_1, a_1, \ldots, o_T, a_T, o_{T+1} \right)$.
> Here $ T $ denotes the final time step before reaching the terminal observation $o_{T+1} $.
>
> The trajectory terminates when the search process either:
> - Reaches the maximum allowed depth, or
> - Encounters a game victory or death.
>
> In our method, we specifically select trajectories that lead to **game death** to generate reflections.
>
>
> **In-Trial Memory:** The in-trial memory is a sequence of recent observation-action pairs within a small time window in the current trial. It is defined as: $ \left( o_{t-1}, a_{t-1}, o_t \right)$ .
> This sequence represents the immediate history leading up to the current state at time t.
>
>
> **Cross-Trial Memory:** The cross-trial memory consists of reflections from previous trials, up to a maximum of $k$ reflections. In our experiments, we set $ k = 3 $.  A reflection is a brief statement summarizing the reason for the previous rollout's failure, such as: "Moving down into the dark place without a light source or any other items to help navigate."
>
> >**Q2:** Additional baselines could be included in the experimental results...
>
> **A2:** Thank you for your suggestions. We have included these baselines into our experiments. For MPRC-DQN, RC-DQN, and Bike+CBR, we use similar settings to ensure a fair comparison. However, the LDTs baseline operates under a different setup as it requires an additional input—a Goal—at each step. This Goal is learned from offline trajectories collected from the environment. The direct comparison with LDTs would therefore be inequitable.
>
> The table compares our MC-DML approach with selected baselines, excluding the game ztuu due to infinite reward loop issues reported in baseline papers. Our method outperforms MPRC-DQN and RC-DQN across all games, with significant gains in challenging games like zork1 and deephome, and achieves state-of-the-art results on 5 games compared to BiKE+CBR. We will include these baselines in the revised version of the paper.
>
> | Game       | MPRC-DQN | RC-DQN | BiKE + CBR | MC-DML |
> |------------|----------|--------|------------|--------|
> | zork1      | 38.3     | 38.8   | 44.3       | 48.66  |
> | deephome   | 1        | 1      | 1          | 67     |
> | Ludicorp   | 19.7     | 17     | 23.8       | 19.67  |
> | Pentari    | 44.4     | 43.8   | 52.1       | 70     |
> | Detective  | 317.7    | 291.3  | 326.1      | 346.67 |
> | Library    | 17.7     | 18.1   | 22.3       | 21     |
> | Balances   | 10       | 10     | 11.9       | 10     |
> | Temple     | 8        | 8      | 7.8        | 8      |
>
>
> > **Q3:** Do you take into account the next step rewards (when present) during the MCTS? If so, how? If not, why not?
>
> **A3:** Yes, the next-step rewards are taken into account during the MCTS. Specifically, the algorithm calculates the cumulative reward by combining the immediate reward from the current action and the discounted rewards from future steps. These cumulative rewards are then used to update the Q-value of the action node, which is computed as a softmax-weighted sum of accumulated rewards.
>
> > **Q4:** ...Considering the input limitations of LLMs, do you compress the trajectories?
>
> **A4:** Thank you for your question. In the case of in-trial memory, which represents a very short time window, it naturally stays within the context length limit as it includes only the most recent observations and actions. However, trajectories, which consist of sequences of state-action pairs over time, can become lengthy. To ensure they fit within the token limit, we compress them when necessary by using a truncation function. If the trajectory exceeds the maximum token limit, the function discards the earliest information while retaining the most recent information.

---

> > ### Author Response · Authors · 2024-11-22
> > **Response to Review [Part 2/2]**
> >
> > > **Q5:** Do you include the initial description of the game as part of the state? Often the opening message of the game will give valuable information about the goal/objective of the game.
> >
> > **A5:** Thank you for your questions. We avoid including game-specific information in the LLM prompts. In some games, the initial game state includes the game name and copyright-related descriptions (such as the publication year and author), but these are discarded after two steps due to the limited time window of our in-trial memory, which is set to two steps. In our baselines, we included both an LLM agent and a reflection agent and observed that neither was able to complete the game effectively.
> >
> >
> > > **Q6:** When doing MCTS rollouts/simulations, how do you sample actions? Is it random or do you use an informed process during simulation as well?
> >
> > **A6:** Thank you for your questions. During the MCTS process, actions in the selection phase are chosen based on probabilities generated by the LLM. However, in the rollouts/simulations phase, actions are sampled randomly.
> >
> > > **Q7:** Why did you select these 9 Jericho games? Could you report performance across all 33 games in the appendix?
> >
> > **A7:** Thank you for your question. We followed the setup of MC-LAVE-RL and selected the same set of games to ensure a fair and consistent comparison. We believe that this selection provides a representative evaluation of our approach.

---

> > ### Comment · Reviewer_djd9 · 2024-11-27
> > **acknowledgement**
> >
> > Thank you for answering all my questions. Answers to questions 1, 2, 3, 4, and 6 should be added to the paper for better clarity.

---

> > > ### Author Response · Authors · 2024-11-30
> > > **Thank you for your advice.**
> > >
> > > Thank you for your feedback. We appreciate your suggestions and will ensure they are reflected in the revised version of the paper.

---

### Official Review · Reviewer_jbdx · 2024-11-04

**Soundness:** 4
**Presentation:** 4
**Contribution:** 4
**Rating:** 8
**Confidence:** 5

**Summary:**

This is a neat paper combining well known MCTS with LLM memory. More specifically, MCTS will be guided by past failure trajectories as learned by the LLM. The idea is demonstrated in a number of text games.

**Strengths:**

This is a neat and effective idea. It may also apply to other domains such as robot planning or math problem solving. It is still not clear how the formalism is incorporated into POMDP.

**Weaknesses:**

The method requires the LLM to remember these failure trajectories. Long term memory can be one limitation, or for complex tasks it may be needed to do some sort of RAG to retrieve the relevant memory, and this can introduce noise.

**Questions:**

- can you apply this idea for other MCTS-RL frameworks?
- can you apply this idea for other tasks which require planning, such as robot task completion, complex math problem solving, etc.?

---

> ### Author Response · Authors · 2024-11-22
> **Response to Reviews**
>
> We appreciate your recognition of our work and your valuable feedback. Below, we provide detailed responses to your questions.
>
> **Q1:** Can you apply this idea for other MCTS-RL frameworks?
>
> >  **A1:** Yes, our approach can be extended to other MCTS-RL frameworks. The core concept lies in leveraging a dynamically memory-guided LLM in conjunction with the exploratory properties of tree search. In our work, the LLM functions as a policy network, guiding MCTS to search more effectively by generating dynamic action probability distributions. This enables the search process to concentrate on the most promising actions.
> Furthermore, the LLM can be adapted to evaluate the value of state nodes, providing value estimates for each node. Incorporating value estimation would improve the accuracy of node evaluations while reducing reliance on extensive simulations. Additionally, within an RL framework, the integration of the LLM and MCTS can facilitate data augmentation, generating more training samples for RL algorithms and further enhancing their performance.
>
>
> **Q2:** Can you apply this idea for other tasks which require planning, such as robot task completion, complex math problem solving, etc.?
>
> > **A2:** Thank you for your question. The MC-DML algorithm we propose is applicable to other planning tasks, especially in environments with large-scale state-action spaces. The core components of our method—the integration of LLMs for heuristic guidance and reflection, and the use of the PUCT algorithm for efficient search—are not limited to natural language inputs or outputs.
>
> > In other planning tasks, states and actions can be represented in structured formats such as key observation sequences and discrete, tokenized actions. These representations can be fed to LLMs for processing and evaluation. Additionally, prompts and memory mechanisms can be tailored to meet the specific requirements of the target tasks. For example, in chess, chess positions and moves can be converted into textual descriptions, allowing effective policies to be learned from them [1]. Similarly, in robotic planning, LLMs can assist in understanding complex instructions or environmental descriptions, while MCTS is used to explore optimal action sequences.
>
> > We chose text-based games as our environments primarily because of their challenges, such as sparse rewards and the need for long-term planning. Future research could explore the potential of MC-DML across diverse planning tasks and RL environments beyond text-based games.
>
> [1] Feng X, Luo Y, Wang Z, et al. ChessGPT: bridging policy learning and language modeling[C]//Proceedings of the 37th International Conference on Neural Information Processing Systems. 2023: 7216-7262.

---

> > ### Comment · Reviewer_jbdx · 2024-11-26
> >
> > Thanks, I am keeping my rating.

---

### Official Review · Reviewer_ZBKR · 2024-11-04

**Soundness:** 3
**Presentation:** 3
**Contribution:** 2
**Rating:** 5
**Confidence:** 4

**Summary:**

This paper introduces Monte Carlo planning with Dynamic Memory-guided Large language model (MC-DML) algorithm, which targets at bringing the language model understanding and reasoning capabilities into Monte Carlo Tree Search and RL. Specifically, the authors propose to add in-trial (which includes exploration and results in the current game), and cross-trial memory (which is reflections from previous experiences) into predictor UCT. Experimenting on 9 text-based games in Jericho benchmark, results show that the proposed MC-DML approach improve the performance over other RL-based, LLM-based, and MCTS-based approaches.

**Strengths:**

1. This paper shows that adding LLM reasoning and planning capabilities, especially when augmenting with cross-trial reflections, can improve exploration performance in MCTS. This can be an interesting direction to study and may suggest good research results to people interested in how to make use of LLMs in RL exploration.
2. This paper compares to previous RL-based, LLM-based, and MCTS-based methods, and shows that when combining LLM with MCTS RL methods, we would benefit from stronger reasoning and planning capabilities.

**Weaknesses:**

1. The main contribution in the proposed algorithm actually lies in introducing cross-trial reflection, as LLMs have been studies in the MCTS setup which is also mentioned by the authors. However, although the paper compares to LLM-based and MCTS-based baselines, there is no comparison to LLM + MCTS baselines, which seems to be the most relevant.
2. Given that combining LLMs and MCTS has been well studies in both text-based games and other RL setups, the contribution of this paper might be relatively limited.

**Questions:**

1. Have you done ablation studies to study the impact of choosing k in the cross-trial memory? Why would 3 reflection be sufficient?
2. How would the proposed method work in other RL setups beyond text-based games?
3. How would the quality of the LLM (for both action prediction, and reflection) impact the overall performance?

---

> ### Author Response · Authors · 2024-11-22
> **Response to Review [Part 1/2]**
>
> **Q1:**  There is no comparison to LLM + MCTS baselines, which seems to be the most relevant.
>
> > **A1:**  Thank you for your question. In our ablation study, we evaluated  the performance of MC-DML without cross-memory (refer to Table 3, "MC-DML w.o. Mc​, DP"), which aligns with the setup of the LLM-MCTS approach [1]. While LLM-MCTS demonstrates strong performance in commonsense reasoning tasks, our experimental results indicate that MC-DML achieves superior performance in text-based games.
>
> **Q2:**  Given that combining LLMs and MCTS has been well studied in both text-based games and other RL setups, the contribution of this paper might be relatively limited.
> > **A2:**  Thank you for raising this point. While combining LLMs and MCTS has been explored in previous studies, our work offers a distinct contribution. We appreciate the opportunity to elaborate on the distinctions.
>
> > **Environment Complexity:** Previous studies, such as Tree of Thought [2], Reasoning via Planning [3], and Language Agent Tree Search [4], primarily focus on relatively simple environments like Game of 24 and Blocksworld. In contrast, our study addresses environments with larger state spaces and highly branched structures. For instance, in the Zork1 game, the state space undergoes combinatorial explosion due to factors such as player location and task progress. The valid actions in each state can reach up to dozens. These environments demand both extensive exploration and long-term reasoning.
>
> > **Methodological Differences:** Previous studies have utilized LLMs as agents to iteratively generate reasoning paths, leveraging search-based methods to navigate these paths. While effective in simpler environments, this approach falls short in addressing the complexity of the environments in our study. LLMs often struggle to align their generated reasoning with executable actions and lack the ability to effectively balance exploration and exploitation. Even when provided with predefined executable actions, their performance remains constrained (as in our baseline ''LLM agent''). In contrast, our framework incorporates the PUCT algorithm and integrates LLMs as a prior policy to guide the search process, enhancing its capability to address these challenges.
>
> **Q3:**  Have you done ablation studies to study the impact of choosing k in the cross-trial memory? Why would 3 reflections be sufficient?
> > **A3:**  Thank you for your question. In our experiments, we observed that the choice of k is closely linked to the complexity of the game. For games with multiple traps, a larger k may be required, whereas simpler games can perform well with a smaller k. Rather than fine-tuning k for each game, we adopted the setup from Reflection [4], selecting k=3. This approach strikes a balance by enabling multiple reflections while maintaining manageable experimental costs.
>
> **Q4:**  How would the proposed method work in other RL setups beyond text-based games?
> > **A4:** Thank you for your question. The MC-DML algorithm we propose is applicable to other RL setups, especially in environments with large-scale state-action spaces. The core components of our method—the integration of LLMs for heuristic guidance and reflection, and the use of the PUCT algorithm for efficient search—are not limited to natural language inputs or outputs. In other RL environments, states and actions can be represented in structured formats such as key observation sequences and discrete, tokenized actions. These representations can be fed to LLMs for processing and evaluation. Similarly, in robotic planning, LLMs can assist in understanding complex instructions or environmental descriptions, while MCTS is used to explore optimal action sequences. Future research can explore the capabilities of MC-DML in a variety of RL environments beyond text-based games.
>
> **Q5:**  How would the quality of the LLM (for both action prediction, and reflection) impact the overall performance?
>
> > **A5:** Thank you for your question. We acknowledge that the in-context learning capabilities of different LLMs can affect action prediction and reflection. Existing studies have extensively explored the scaling laws in this area. We utilized GPT-3.5 as the backend model for our MC-DML approach in the experiments to demonstrate its effectiveness. This choice aligns with prior work such as LLM-MCTS [1], which also relied exclusively on the GPT-3.5 model.

---

> > ### Comment · Reviewer_ZBKR · 2024-11-24
> > **Thanks for the response and clarification.**
> >
> > 1. Thanks for pointing out the LLM + MCTS results. First of all, LLM + MCTS as I explained, should be THE MAIN baseline to compare to, rather than one of the ablation studies. Moreover, the ablation you showed coupled DP with MC. Why is there no baseline of LLM + MCTS directly?
> > 2. Regarding k, my original question was suggesting whether a higher k would improve the performance (given that only k = 3 was used), especially with regarding to your argument in question 2 where the games studied are much more complex.
> > 3. Regarding the quality of LLMs, because of the approach proposed, it seems that it would be more sensitive to the model quality. I was suggesting that it would be more interesting to add some analysis on this perspective, regardless of what models previous papers used (since their approach might be less sensitive, so this question is less relevant)

---

> > > ### Author Response · Authors · 2024-11-30
> > > **Response to Review**
> > >
> > > We appreciate your prompt feedback. To address your remaining concerns, we provid point-by-point responses below.
> > >
> > > > **Q1:** Regarding the LLM + MCTS baseline.
> > >
> > > **A1:**  We agree that LLM + MCTS serves not only as an ablation study but also as a key baseline. We initially categorized it under ablation studies for two reasons:
> > > - To the best of our knowledge, there has been no prior implementation of LLM + MCTS in text-based games. While Zhao et al. (2023) proposed LLM + MCTS for object rearrangement tasks, our implementation in text-based games differs in several aspects. For instance, text-based games feature a fixed initial node, whereas Zhao et al. (2023)  leveraged LLMs as a world model to infer the starting position.
> > > - Our approach applies LLM + MCTS to text-based games and further introduces MC-DML. To highlight the importance of the cross-memory component, we conducted ablation experiments to assess its impact.
> > >
> > > Following your suggestion, we includ a new baseline experiment, **MCTS + LLM (vanilla)**, also referred to as **MC-DML w/o $M_c$**, where we only remove the cross-memory component from MC-DML but keep dynamic pruning.
> > >
> > > | Game        | MC-DML          | MCTS + LLM (vanilla) |
> > > |-------------|------------------|--------------------|
> > > | Zork1       | 48.66 ± 1.89    | 38.33 ± 2.89         |
> > > | Deephome    | 67 ± 1.41       | 62.66 ± 0.94      |
> > > | Detective   | 346.67 ± 9.43   | 326.67 ± 4.71     |
> > > | Ztuu        | 23.67 ± 1.9     | 20.66 ± 0.47      |
> > >
> > > In our revised manuscript, we will emphasize this model within both the baseline and the ablation studies sections.
> > >
> > > > **Q2:** Regarding the $k$.
> > >
> > > **A2:** We conducted experiments on Zork1 and Ludicorp games with $k$ increased to 5, but observed no significant performance improvement:
> > >
> > > | Game     | k=3         | k=5         |
> > > |----------|-------------|-------------|
> > > | Zork1    | 48.66 ± 1.89| 48 ± 1.63   |
> > > | Ludicorp | 67 ± 1.41   | 68 ± 1.63   |
> > >
> > > This might be due to the fact that in our experiments, the trajectory used for reflection begins from the current search's root node, rather than from the start of the trajectory. The length of this segment is constrained by the maximum search depth of MCTS. Despite the complexity of the games, the traps encountered in these shorter segments are limited, which is influenced by the game's design. We will include these details in the appendix of the revised version.
> > >
> > >
> > > > **Q3:**  Regarding the quality of different LLMs.
> > >
> > > **A3:**  Thank you for your question. We acknowledge that the quality of different LLMs and their impact on our proposed approach is an important factor to consider. However, this lies beyond the scope of our current study and will be left for future research.

---

> > > > ### Author Response · Authors · 2024-12-03
> > > > **Thank you for your feedback**
> > > >
> > > > Thank you for your feedback, which has helped improve our paper.
> > > >
> > > > We have provided detailed responses to address your concerns. As the discussion period is ending, we kindly ask if you have any remaining questions about our work.

---

> ### Author Response · Authors · 2024-11-22
> **Response to Review [Part 2/2]**
>
> References
>
> > [1] Zhao Z, Lee W S, Hsu D. Large language models as commonsense knowledge for large-scale task planning[C]//Proceedings of the 37th International Conference on Neural Information Processing Systems. 2023: 31967-31987.
>
> > [2] Yao S, Yu D, Zhao J, et al. Tree of thoughts: deliberate problem solving with large language models[C]//Proceedings of the 37th International Conference on Neural Information Processing Systems. 2023: 11809-11822.
>
> > [3] Hao S, Gu Y, Ma H, et al. Reasoning with language model is planning with world model[J]. arXiv preprint arXiv:2305.14992, 2023.
> Zhou A, Yan K, Shlapentokh-Rothman M, et al. Language agent tree search unifies reasoning acting and planning in language models[J]. arXiv preprint arXiv:2310.04406, 2023.
>
> > [4] Shinn N, Cassano F, Gopinath A, et al. Reflexion: language agents with verbal reinforcement learning[C]//Thirty-seventh Conference on Neural Information Processing Systems.

---

### Meta-Review · Area_Chair_BvWP · 2024-12-19

**Metareview:**

The paper presents a more efficient way to do Monte Carlo planning in text games. It is well situated with respect to the literature on text game playing agents, especially those that already attempt to solve them via a mix of Deep RL and search+backtracking methods. The addition of LLMs has always been bottlenecked by the compute costs but the MC-DML could be a good answer to that. The results are also well presented and show a solid improvement over existing methods. Some of the exact implementation details are unclear from the paper itself and I'd encourage the authors to make the code for this work available if they can as that will allow for other papers to build on it - implementation details are critical for such methods especially.

**Additional Comments On Reviewer Discussion:**

There seems to have been reasonable discussion on the points raised by the reviewers. Most of them revolved around additional clarifications or implementation details to which the authors responded sufficiently.

---

### Decision · Program_Chairs · 2025-01-22

Accept (Poster)